# Immune Checkpoint Inhibitor Therapy Induces Inflammatory Activity in the Large Arteries of Lymphoma Patients under 50 Years of Age

**DOI:** 10.3390/biology10111206

**Published:** 2021-11-19

**Authors:** Raffaella Calabretta, Philipp B. Staber, Christoph Kornauth, Xia Lu, Patrick Binder, Verena Pichler, Markus Mitterhauser, Alexander Haug, Xiang Li, Marcus Hacker

**Affiliations:** 1Division of Nuclear Medicine, Department of Biomedical Imaging and Image-Guided Therapy, Medical University of Vienna, 1090 Vienna, Austria; raffaella.calabretta@meduniwien.ac.at (R.C.); lxgf2222@163.com (X.L.); patrick.binder@meduniwien.ac.at (P.B.); verena.pichler@univie.ac.at (V.P.); markus.mitterhauser@meduniwien.ac.at (M.M.); alexander.haug@meduniwien.ac.at (A.H.); xiang.li@meduniwien.ac.at (X.L.); 2Division of Hematology, Department of Medicine I, Medical University of Vienna, 1090 Vienna, Austria; philipp.staber@meduniwien.ac.at (P.B.S.); christoph.kornauth@meduniwien.ac.at (C.K.)

**Keywords:** atherosclerosis, cardiovascular toxicity, cardio-oncology, immune checkpoint inhibitor, PET

## Abstract

**Simple Summary:**

Immune checkpoint inhibitor (ICI) therapy has changed the management of many cancers endowed with poor prognosis. However, cardiotoxicity, as well as the possible progression of atherosclerosis, have been described. 2-[18F]fluorodeoxyglucose (FDG) positron emission tomography (PET) is a validated tool to quantify atherosclerotic inflammatory activity; therefore, we found it interesting to analyze the changes in maximum FDG standardized uptake values (SUV_max_) and of target-to-background ratios (TBRs) in 117 arterial segments of 12 otherwise healthy, young lymphoma patients, underwent PET pre/post ICI treatment. As systemic immune activation surrogate markers, SUV_max_ of the bone marrow, spleen, and liver, as well high-sensitivity C-reactive protein (hsCRP) pre- and post-treatment, were additionally analyzed. ICI therapy induced arterial inflammatory activity, detected by increased TBR in all PET lesions. FDG uptake measured in other organs and hsCRP levels remained unchanged. Our findings show that cancer immunotherapy with ICI might be a critical moderator of atherosclerosis, with a possible subsequently increased risk of future cardiovascular events in oncological patients, even in young patients with low cardiovascular risk.

**Abstract:**

**Background:** Immune checkpoint inhibitors (ICI) have transformed the management of various cancers. Serious and potentially fatal cardiovascular toxicity, as well as a progression of atherosclerosis, have been described, mainly in elderly and comorbid patients. **Methods:** We investigated 117 arterial segments of 12 young (under 50 years of age), otherwise healthy lymphoma patients pre/post-ICI treatment using 2-[18F]fluorodeoxyglucose (FDG) positron emission tomography (PET). Maximum FDG standardized uptake values (SUV_max_) and target-to-background ratios (TBRs) were calculated along arterial segments. Additionally, metabolic activities (SUV_max_) of the bone marrow, spleen, and liver were analyzed. The levels of high-sensitivity C-reactive protein (hsCRP) were assessed. **Results:** ICI therapy induced arterial inflammatory activity, detected by increased TBR in arterial segments without pre-existing inflammation (TBRneg_pre = 1.20 ± 0.22 vs. TBRneg_post = 1.71 ± 0.45, *p* < 0.001), whereas already-inflamed lesions remained unchanged. Dormant calcified segments (Hounsfield Units-HU ≥ 130) showed a significant increase in TBR values after ICI treatment (TBRcalc_pre = 1.36 ± 0.38 vs. TBRcalc_post = 1.76 ± 0.42, *p* < 0.001). FDG uptake measured in other organs and hsCRP levels remained unchanged after ICI therapy. **Conclusions:** Although the effects of ICI therapy on arterial inflammation are still incompletely understood, cancer immunotherapy might be a critical moderator of atherosclerosis with a subsequently increased risk of future cerebro- and/or cardiovascular events in young oncological patients.

## 1. Introduction

Targeted therapies are amongst the major treatment options for cancer today. Immune checkpoint inhibitors (ICI) targeting programmed death-1/ligand-1 (PD-1/PD-L1) have transformed the management of different prognostically poor cancers [1,2,3]. Interferences with the PD-1 axes can cause an activation of autoreactive T-cells, damaging host tissues, and different immune-related adverse events (irAEs) have previously been reported. The known irAEs may affect a variety of organs such as the bowel, thyroid, liver, pituitary gland, and musculoskeletal system. Potential cardiovascular and cardiotoxicity have also been described [1,4,5]. PD-1/PD-L1 chemokine axes could play important roles in limiting T-cell-mediated autoimmune inflammation. Previous preclinical studies have suggested that activated T-cells produce large amounts of pro-atherogenic cytokines, potentially contributing to both the growth and destabilization of atherosclerotic plaques [6]. Positron emission tomography (PET) with 2-[18F]fluorodeoxyglucose (FDG) is a validated tool to assess atherosclerotic inflammatory activity, including in cancer patients [7,8]. Recently, we described an elevated inflammation determined by increased FDG-uptake in the large arteries after immunotherapy in older patients (mean age: 71 ± 14 years) suffering from melanoma and treated with ICI, suggesting increased atherosclerotic inflammation [9]. There is less published knowledge regarding how ICI affects arterial inflammation in younger patients; therefore, in this present study, we aimed to analyze the cardiovascular toxicity induced by ICI treatment in patients with lymphoma under 50 years of age with low cardiovascular risk.

## 2. Materials and Methods

Twelve lymphoma patients (seven females, five males, mean age: 35 ± 9 years) treated with PD-1 inhibitors who underwent FDG positron emission tomography–computer tomography (PET/CT) or positron emission tomography–magnetic resonance imaging (PET/MRI) scans performed for diagnostic purposes before and after treatment (time interval between pre- and post-scans: 9.6 ± 3.9 months) were retrospectively analyzed. The study was conducted according to the guidelines of the Declaration of Helsinki, and approved by the Ethics Committee of the University of Vienna (approval no. 1367/2020). Baseline patient characteristics, as well their cardiovascular risk factors and previous cancer treatments, were recorded (Table 1). Maximum FDG standardized uptake values (SUV_max_) were corrected for FDG blood-pool activity (SUV_bloodpool_) to calculate target-to-background ratios (TBRs), as previously described [10].

We analyzed 117 arterial lesions of 6 arterial segments (ascending and descending aorta, aortic arch, abdominal aorta, and iliac arteries) in PET scans pre- and post-ICI treatment, and quantified segmental FDG accumulation according to the current guidelines [10]. We classified arterial lesions by pre-existing active inflammation (cut-off: TBRpre ≥ 1.48). Segmental calcium density categorization as non-calcified (<130 Hounsfield Units-HU) or calcified (≥130 HU) was only possible in eight patients who received PET/CT before and after ICI treatment. Continuous variables were recorded as the mean ± standard deviation. The mean FDG uptake values (SUV_max_ and TBR_max_) pre- and post-therapy were retrospectively assessed using the paired Student’s t-test. The change in TBR values (ΔTBR = TBR_post_ − TBR_pre_) was subsequently calculated and compared using the ANOVA test. Two-sided *p*-values of <0.05 were considered significant.

As surrogate markers for systemic immune activation, FDG uptake as SUV_max_ was measured in bone marrow and spleen before and after ICI therapy, as previous described [9]. The hepatic activity also as SUV_max_ was additionally measured by manually placing three regions of interest (ROIs) in coronal, axial, and longitudinal projections of the organ parenchyma, and liver-to-spleen ratios (normally >1) were subsequently calculated [11]. Patients with splenic and/or hepatic metastases were excluded from this analysis.

As an additional systemic inflammation marker, high-sensitivity C-reactive protein (hsCRP) levels before and after ICI therapy were also collected.

## 3. Results

ICI immunotherapy resulted in significant increases in the inflammatory activity in all assessed arterial PET lesions (*n* = 117, lesional TBR_pre_ = 1.50 ± 0.42 vs. lesional TBR_post_ = 1.79 ± 0.46, *p* < 0.001) (Figure 1A). A significant increase in lesional calcification after therapy was also found (HU_pre_ = 147 ± 59 vs. HU_post_ = 177 ± 65, *p* < 0.001). Significant increases in TBR were found in lesions without pre-existing arterial inflammation (*n* = 65, TBR_neg_pre_ = 1.20 ± 0.22 vs. TBR_neg_post_ = 1.71 ± 0.45, *p* < 0.001), whereas lesions with pre-existing active inflammation remained largely unchanged (*n* = 52, TBR_pos_pre_ = 1.85 ± 0.36 vs. TBR_pos_post_ = 1.87 ± 0.51, *p* = 0.834). In contrast, in calcified lesions, TBR values were significantly elevated after therapy (*n* = 73, TBR_calc_pre_ = 1.36 ± 0.38 vs. TBR_calc_post_ = 1.76 ± 0.42, *p* < 0.001), whereas no significant change was observed in non-calcified PET lesions (*n* = 14, TBR_non_calc_pre_ = 1.97 ± 0.35 vs. TBR_non_calc_post_ = 1.92 ± 0.25, *p*= 0.683).

To derive a deeper insight into the relationship between pre-existing inflammation, calcification and increased inflammation after treatment, we generated four subgroups of arterial segments by calcification (yes/no) and pre-existing inflammation (yes/no). Calcified lesions without active inflammation showed a significant increase in inflammation after ICI, regardless of the severity of calcification (*n* = 47, TBR_calc_neg_pre_ = 1.16 ± 0.18 vs. TBR_calc_neg_post_ = 1.74 ± 0.46, *p* < 0.001). In contrast, no significant changes in inflammation were found in initially inflamed lesions, independent of pre-existing calcification (Figure 1B).

FDG uptake measured in bone marrow, spleen, and liver did not show statistical differences before and after ICI (SUV_max_bonemarrow_pre_ = 1.97 ± 0.85 vs. SUV_max_bonemarrow_post_ = 1.86 ± 0.84, *p* = 0.075; SUV_max_spleen_pre_ = 3.77 ± 2.17 vs. SUV_max_spleen_post_ = 3.05 ± 1.02, *p* = 0.343; SUV_max_liver_pre_ = 3.49 ± 0.94 vs. SUV_max_liver_post_ = 3.18 ± 0.61, *p* = 0.197).

There were also no significant differences in the spleen-to-liver ratios measured in pre- and post-FDG PET scans (S/L_pre_ = 2.11 ± 0.44 vs. S/L_post_ = 2.13 ± 0.50, *p* = 0.889). The inflammatory blood biomarker hsCRP markedly increased after therapy, but without statistical significance (hsCRP_pre_ = 2.66 ± 4.61 vs. hsCRP_post_ = 6.96 ± 15.77; *p* = 0.393).

## 4. Discussion

The present study demonstrates a significant effect of ICI treatment on arterial inflammation in lymphoma patients under 50 years of age, without high cardiovascular risk. The underlying mechanism of inflammatory activation by ICI treatment, however, remains unclear. Pioneering studies have demonstrated that PD-L1 is expressed on endothelial cells and antigen-presenting cells (such as dendritic cells or macrophages) in the vasculature, whereas PD-1 is expressed on activated T-cells [12,13]. The protective immunity of vascular PD-L1 and T-cell PD-1 interaction downregulates the T-cell activation coupled with subsequent apoptosis or the suppression of cluster of differentiation 4 (CD4+) and CD8+ T-cell proliferation [13]. Hence, inhibiting the PD-1 pathway could induce vascular inflammation by ICI treatment, which was observed in our findings. Nonetheless, we found that ICI therapy selectively activates low-active lesions, but not pre-active lesions. It could be speculated that PD-L1 blockade could compromise vascular integrity in early atherosclerosis by allowing activated PD-1^high^ T-cells to interact with the PD-L1^high^ endothelium/macrophages. Furthermore, the upregulation of vascular inflammation/metabolism was observed in arterial segments without present inflammation. Negligible activation in pre-activated segments by immunotherapy with ICI, probably because the deficient infiltration PD-1^high^ T-cells, induces pre-vascular inflammation, which disrupts the protective effect of the PD-L1 and PD-1 axes. We also hypothesize that previous anti-cancer therapies might induce the pre-vascular inflammation in respect to endothelial/macrophages activation, which hampers the pro-inflammation modulation of subsequent ICI treatment.

Taken together, this pre-existing unbalanced axis between PD-L1^high^ macrophages/endothelial cells and PD-1^high^ T-cells in inflamed arterial segments delivers a lower sensitivity to ICI therapy. In line with our hypothesis, this could be related to previous increases in effector-memory CD4+ T-cells in these inflamed arterial segments [14]. In summary, the vascular PD-L1 and PD-1 axes could exert an atheroprotective mechanism to maintain peripheral immune tolerance, and an intercepted PD-1/PD-L1 axis between PD-1^high^ T-cells and PD-L1^high^ endothelium/macrophages could destruct vascular integrity [12,15]. PD-1 deficiency or inhibition of the T-cells could accelerate intimal infiltration of CD8+ T-cells without directly affecting the myeloid system, according with our findings regarding non-statistical significance changes in the uptake of spleen and bone marrow [14].

The main limitations of this preliminary report are the very small patient population and the lack of follow-up data concerning the clinical outcome of the patients.

A long-term follow-up of the described subjects is currently ongoing. However, it is significant to consider that many of these patients have an advanced cancer disease and may not complete the follow-up as planned due to premature death.

Furthermore, we are also working on other larger patient cohorts including subjects affected by other malignancies. To provide further insights, prospective studies on oncological patients receiving ICI should be performed. In these next analysis, patients cardiovascular risk profile should be described in more detail, including through more specific blood tests and adequate follow-ups. Finally, a study should be conducted also in atherosclerosis preclinical models, in order to obtain more information in regard to the mechanism behind this treatment and to enlighten the potential toxic side effects of ICI treatment.

## 5. Conclusions

The consequences of ICI therapy on atherosclerosis are still incompletely understood.

The early identification of patients at risk of cardiovascular and cardiotoxicity from cancer target therapies, as well the early diagnosis of possible cardiovascular complications after ICI exposure, are crucial.

Our findings suggested that cancer ICI immunotherapy could be a critical regulator of atherosclerosis, with a possible increased risk of future cerebro- and/or cardiovascular events in younger patients with low cardiovascular risk and longer life expectancy.

## Figures and Tables

**Figure 1 biology-10-01206-f001:**
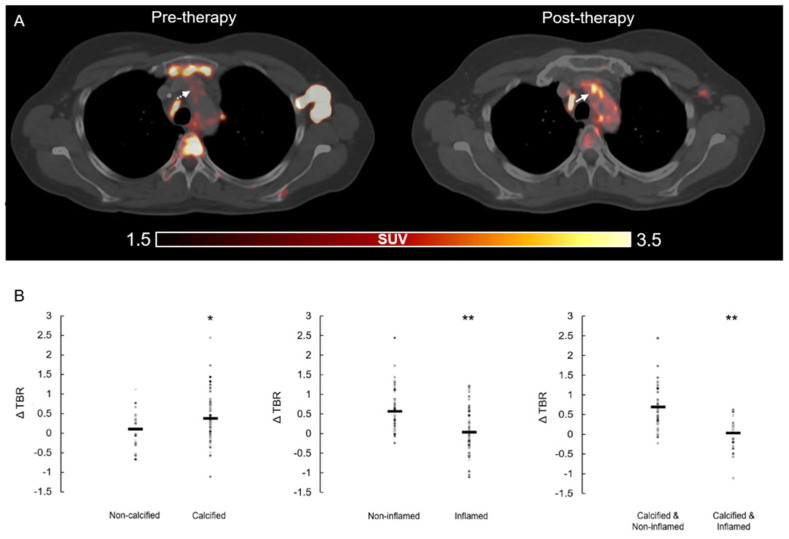
(**A**). Representative PET/CT images. Increased arterial FDG uptake (white arrow) after ICI therapy compared to baseline examination (pre-therapy). PET, positron emission tomography; CT, computer tomography; FDG, 2-[^18^F]fluorodeoxyglucose; ICI, immune checkpoint inhibitors. (**B**). Change in inflammatory activity (ΔTBR) in arterial segments. Given four subgroups with/without calcification as well as pre-existing/absent active inflammation (cut-off: TBRpre = 1.48). TBR, target-to-background ratio. * Significance of *p* < 0.05, ** significance of *p* < 0.001.

**Table 1 biology-10-01206-t001:** Baseline patient characteristics. General features of the patient population.

Gender (males/females)	5/7
Age (years)	35 ± 9
Diagnosis HL, N (%)	7 (58)
Diagnosis NHL, N (%)	5 (42)
ICI Therapy (N; %)	PD-1 Inhibitors (12; 100) ➢ Pembrolizumab (7; 58) ➢ Nivolumab (5; 42)
Previous ICI Therapy, N (%)	2 (17)
Following ICI Therapy, N (%)	1 (8)
CHT before ICI therapy, N (%)	12 (100)
RT during ICI therapy, N (%)	1 (8)
RT before ICI therapy, N (%)	3 (25)
BMI (Kg/m2)	25 ± 6
Smoking, N (%)	2 (17)
Hypertension, N (%)	0 (0)
Dyslipidemia, N (%)	0 (0)
Diabetes, N (%)	1 (8)
Prior myocardial infarction, N (%)	0 (0)
Prior TIA/Stroke, N (%)	0 (0)
PAD, N (%)	1 (8)

Abbreviations. HL: Hodgkin lymphoma; NHL: non-Hodgkin lymphoma; ICI: immune checkpoint inhibitor; CHT: chemotherapy; RT: radiotherapy; BMI: body mass index; TIA: transient ischemic attack; PAD: peripheral artery disease.

## Data Availability

Not applicable.

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
