# Peer review of "Immune Checkpoint Inhibitor Therapy Induces Inflammatory Activity in the Large Arteries of Lymphoma Patients under 50 Years of Age"

_biology, 2021, doi:10.3390/biology10111206_

Round 1

Reviewer 1 Report

In the present study, Calabretta and colleagues investigated an impact of immune checkpoint inhibitor therapy with PD-1 inhibitors (pembrolizumab/ nivolumab) on atherosclerosis and cardiovascular events in 12 lymphoma patients (< 50 years) by diagnostic imaging procedures via 18FDG-PET/CT/MRI and quantification of C-reactive protein as an indicator of inflammation. Signs of arterial inflammation were detected in arterial segments without previous inflammation as well as calcified segments. The authors' findings support previous data on the induction of chronic inflammation as a precursor to atherosclerotic cardiovascular disease by the increasing use of immune checkpoint inhibitors in oncology as a high-risk factor for such adverse events in long-term cancer survivors. However, the risk described in this paper is already well known and the method of 18FDG-PET CT/MRI is an established diagnostic tool in this context. The authors describe the phenomenon in another very small, albeit younger, cohort but do not provide any mechanistic information on the underlying pathophysiological pathways. Nevertheless, the study is well conducted and the results are clearly described for a tumor entity not previously studied in this regard, at least to my knowledge.

Just some minor comments:

  1. Line 195: The authors should indicate whether they plan to study a larger cohort and whether a long-term follow-up of the patients described here is performed.
  2. Apart from the known facts, what conclusions do the authors draw from their study? this point comes up short and requires further elaboration.

Author Response

Reviewer: 1
Comments and Suggestions for Authors: In the present study, Calabretta and colleagues investigated an impact of immune checkpoint inhibitor therapy with PD-1 inhibitors (pembrolizumab/ nivolumab) on atherosclerosis and cardiovascular events in 12 lymphoma patients (< 50 years) by diagnostic imaging procedures via 18FDG-PET/CT/MRI and quantification of C-reactive protein as an indicator of inflammation. Signs of arterial inflammation were detected in arterial segments without previous inflammation as well as calcified segments. The authors' findings support previous data on the induction of chronic inflammation as a precursor to atherosclerotic cardiovascular disease by the increasing use of immune checkpoint inhibitors in oncology as a high-risk factor for such adverse events in long-term cancer survivors. However, the risk described in this paper is already well known and the method of 18FDG-PET CT/MRI is an established diagnostic tool in this context. The authors describe the phenomenon in another very small, albeit younger, cohort but do not provide any mechanistic information on the underlying pathophysiological pathways. Nevertheless, the study is well conducted and the results are clearly described for a tumor entity not previously studied in this regard, at least to my knowledge. Just some minor comments:

Comment 1: Line 195: The authors should indicate whether they plan to study a larger cohort and whether a long-term follow-up of the patients described here is performed.

Response: We thank the Reviewer for his/her comment. Currently, a long-term follow-up of the patients described is ongoing and we conducting animal experiments in order to obtain more information in regard the mechanism behind this treatment and to enlighten potential toxic side effects of ICI treatment. We are also working on a larger cohort (about 50 subjects) of patients suffering from lung cancer treated with ICI therapy.

Furthermore, we are already planning a prospective study on oncological patients receiving ICI, where we will document their cardiovascular risk profile in more detail accompanied by more specific blood tests and adequate follow-ups. However, it is significant to consider that many of these patients have advanced cancer disease and may not complete the follow-up as planned due to premature death. However, the early identification of patients at (higher) risk for cardiotoxicity from cancer therapy, it is crucial to detect previous subclinical cardiac abnormalities and to perform an early detection of possible cardiovascular complications during treatment. We add this point in the Discussion (page 7, lines 206 - 215). 

Comment 2: Apart from the known facts, what conclusions do the authors draw from their study? This point comes up short and requires further elaboration.

Response: Thank you for this comment. The early identification of patients at risk for cardiovascular and/or cardiotoxicity from cancer therapy is crucial in order to predict future cerebro- and/or cardiovascular events. As our preliminary results are nicely demonstrating, this is particularly true in younger patients with higher life expectancy. We explained this point in the Conclusion (page 7, lines 220 - 226).

Reviewer 2 Report

Raffaella Calabretta and colleagues present a quality and well-written communication manuscript showing that immune checkpoint inhibitor therapy induces inflammatory activity in large arteries of lymphoma patients below 50 years old.

Authors investigated 117 arterial segments of 12 younger (below an age of 50 years) otherwise healthy lymphoma patients pre/post ICI treatment using 2-[18F]fluorodeoxyglucose positron emission tomography. FDG maximum standardized uptake  values and target-to-background ratios were quantified along arterial segments. Metabolic activities of bone marrow, spleen, and liver were additionally analysed. The levels of high sensitivity C-reactive protein were also collected.

They demonstrated that ICI therapy induced arterial inflammatory activity as detected by increased TBR in arterial segments without pre-existing inflammation, while already inflamed lesions remained unchanged. Dormant calcified segments showed a significant increase of TBR values after ICI treatment. FDG uptake measured in other organs and the hsCRP levels remained unchanged after ICI therapy.

Based on the obtained results authors conclude that although the effects of ICI therapy on arterial inflammation are still incompletely understood, cancer immunotherapy might be a critical moderator of atherosclerosis with subsequent increased risk of future cerebro- and/or cardiovascular events in young oncological patients.

Other comments:

1) Please check for typos throughout the manuscript.

2) Authors are kindly encouraged to cite the following article that describes various aspects of immunotherapies, including immune checkpoint inhibitor therapies. DOI: 10.3390/cancers13040743 (Pubmed ID 33670139)

Overall, the manuscript is valuable for the scientific community and should be accepted for publication after minor corrections are made.

Author Response

Reviewer: 2

Comments and Suggestions for Authors: Raffaella Calabretta and colleagues present a quality and well-written communication manuscript showing that immune checkpoint inhibitor therapy induces inflammatory activity in large arteries of lymphoma patients below 50 years old.

Authors investigated 117 arterial segments of 12 younger (below an age of 50 years) otherwise healthy lymphoma patients pre/post ICI treatment using 2-[18F]fluorodeoxyglucose positron emission tomography. FDG maximum standardized uptake values and target-to-background ratios were quantified along arterial segments. Metabolic activities of bone marrow, spleen, and liver were additionally analysed. The levels of high sensitivity C-reactive protein were also collected.

They demonstrated that ICI therapy induced arterial inflammatory activity as detected by increased TBR in arterial segments without pre-existing inflammation, while already inflamed lesions remained unchanged. Dormant calcified segments showed a significant increase of TBR values after ICI treatment. FDG uptake measured in other organs and the hsCRP levels remained unchanged after ICI therapy.

Based on the obtained results authors conclude that although the effects of ICI therapy on arterial inflammation are still incompletely understood, cancer immunotherapy might be a critical moderator of atherosclerosis with subsequent increased risk of future cerebro- and/or cardiovascular events in young oncological patients. Other comments:

Comment 1: Please check for typos throughout the manuscript.

Response: We thank the Reviewer for this kind remark. We corrected the typos in the manuscript. We revised our article using the “Track Changes” function of Word, as requested.

Comment 2: Authors are kindly encouraged to cite the following article that describes various aspects of immunotherapies, including immune checkpoint inhibitor therapies. DOI: 10.3390/cancers13040743 (Pubmed ID 33670139).

Response: We cited this article in our Introduction (page 3, line 60).  We added this article in the References section as reference no. 3 (page 8, lines 249 - 250).

Comment 3: Overall, the manuscript is valuable for the scientific community and should be accepted for publication after minor corrections are made.

Response: We thank the Reviewer for this comment. We tried our best to improve our manuscript, as suggested.

Reviewer 3 Report

The communication entitled "Immune checkpoint inhibitor therapy induces inflammatory activity in large arteries of Lymphoma patients below 50 years of age" explains the side effect of ICI therapy that includes cardiotoxicity in young patients.

Comments

1. This is a small cohort of patients and they don't have any follow-up data.

2. How will the authors know it's not the effect of previous therapies?

3. Do you have any data on another cancer type that uses ICI therapy?

4. Do you have any data that uses ICI therapy alone.

5. Is it possible to show this toxicity in any preclinical models, since the cohort is so small?

Author Response

Reviewer: 3

Comments and Suggestions for Authors: The communication entitled "Immune checkpoint inhibitor therapy induces inflammatory activity in large arteries of Lymphoma patients below 50 years of age" explains the side effect of ICI therapy that includes cardiotoxicity in young patients. Comments:

Comment 1: This is a small cohort of patient and they do not have any follow-up data.

Response: Thank you.  The present study is conducted retrospectively and we tried our best to collect all available patient information. The long-term follow-up is ongoing. We add this point in the Discussion section (page 7, lines 206 - 215). 

Comment 2:  How will the authors know it's not the effect of previous therapies?

Response: Thank you for this very important comment. Indeed, there were some patients treated with previous RT (3/12), which might have affected arterial inflammation at the baseline scan before ICI, even if the time interval was about 8 weeks minimum between RT and PET pre-ICI. The time interval minimum between CHT and PET pre-ICI was about 6 weeks. However, as we measured the effect of ICI as the change of signal between baseline and follow-up after ICI, the probability of dynamic effects from a previous therapy to our point of view is low.

Comment 3: Do you have any data on another cancer type that uses ICI therapy?

Response: Recently, we analyzed a melanoma patient cohort, older (mean age 71±14 years) compared to the present lymphoma patients (mean age 35±9 years). We described an elevating inflammation determined by increased FDG-uptake in the large arteries after immunotherapy with ICI (Calabretta R, Hoeller C, Pichler V, Mitterhauser M, Karanikas G, Haug A, Li X, Hacker M. Immune Checkpoint Inhibitor Therapy Induces Inflammatory Activity in Large Arteries. Circulation. 2020 Dec 15;142(24):2396-2398).

Furthermore, we are working currently on a larger cohort (about 50 subjects) including patients suffering from lung cancer treated with ICI therapy. Also in this group, we are observing similar results regarding the changes of arterial metabolic activity after immunotherapy.

Comment 4: Do you have any data that uses ICI therapy alone.

Response: No, unfortunately we have no data regarding patients treated only with ICI therapy. In all patient cohort treated with ICI that we analyzed (melanoma, lymphoma, and lung cancer), there are some patients treated with previous chemotherapy and/or with previous radiotherapy.

Comment 5: Is it possible to show this toxicity in any preclinical models, since the cohort is so small?

Response: Thank you for this comment. Poels et al. demonstrated that short-term ICI therapy induces T cell–mediated atherosclerotic plaque inflammation and drives plaque progression in mice (Poels K, van Leent MMT, Boutros C, et al. Immune Checkpoint Inhibitor Therapy Aggravates T Cell–Driven Plaque Inflammation in Atherosclerosis. JACC: CardioOncologyVolume 2, Issue 4, November 2020).

We are currently conducting also a preclinical study in atherosclerosis mouse models and xenografts to get some more mechanistical insights and also to enlighten potential toxic side effects of ICI treatment.